# LightGCL: Simple Yet Effective Graph Contrastive Learning for Recommendation

**Xuheng Cai    Chao Huang**[*]  **Lianghao Xia    Xubin Ren**
Department of Computer Science, University of Hong Kong
{rickcai, lhaoxia}@hku.hk  chaohuang75gmail.com   xubinrencs@gmail.com

## Abstract

Graph neural network (GNN) is a powerful learning approach for graph-based recommender systems. Recently, GNNs integrated with contrastive learning have shown superior performance in recommendation with their data augmentation schemes, aiming at dealing with highly sparse data. Despite their success, most existing graph contrastive learning methods either perform stochastic augmentation (e.g., node/edge perturbation) on the user-item interaction graph, or rely on the heuristic-based augmentation techniques (e.g., user clustering) for generating contrastive views. We argue that these methods cannot well preserve the intrinsic semantic structures and are easily biased by the noise perturbation. In this paper, we propose a simple yet effective graph contrastive learning paradigm LightGCL that mitigates these issues impairing the generality and robustness of CL-based recommenders. Our model exclusively utilizes singular value decomposition for contrastive augmentation, which enables the unconstrained structural refinement with global collaborative relation modeling. Experiments conducted on several benchmark datasets demonstrate the significant improvement in performance of our model over the state-of-the-arts. Further analyses demonstrate the superiority of LightGCL's robustness against data sparsity and popularity bias. The source code of our model is available at https://github.com/HKUDS/LightGCL.

## 1 Introduction

Graph neural networks (GNNs) have shown effectiveness in graph-based recommender systems by extracting local collaborative signals via neighborhood representation aggregation (Wang et al., 2019; Chen et al., 2020b). In general, to learn user and item representations, GNN-based recommenders perform embedding propagation on the user-item interaction graph by stacking multiple message passing layers for exploring high-order connectivity (He et al., 2020; Zhang et al., 2019; Liu et al., 2021a). Most GNN-based collaborative filtering models adhere to the supervised learning paradigm, requiring sufficient quality labelled data for model training. However, many practical recommendation scenarios struggle with the data sparsity issue in learning high-quality user and item representations from limited interaction data (Liu et al., 2021b; Lin et al., 2021). To address the label scarcity issue, the benefits of contrastive learning have been brought into the recommendation for data augmentation (Wu et al., 2021). The main idea of contrastive learning in enhancing the user and item representation is to research the agreement between the generated embedding views by contrasting the defined positive pairs with negative instance counterparts (Xie et al., 2022).

While contrastive learning has been shown to be effective in improving the performance of graph-based recommendation methods, the view generators serve as the core part of data augmentation through identifying accurate contrasting samples. Most of current graph contrastive learning (GCL) approaches employ heuristic-based contrastive view generators to maximize the mutual information between the input positive pairs and push apart negative instances(Wu et al., 2021; Yu et al., 2022a; Xia et al., 2022b). To construct perturbed views, SGL (Wu et al., 2021) has been proposed to generate node pairs of positive view by corrupting the structural information of user-item interaction graph using stochastic augmentation strategies, e.g., node dropping and edge perturbation. To improve the

---

[*]Chao Huang is the corresponding author.

graph contrastive learning in recommendation, SimGCL (Yu et al., 2022a) offers embedding augmentation with random noise perturbation. To work on identifying semantic neighbors of nodes (users and items), HCCF (Xia et al., 2022b) and NCL (Lin et al., 2022) are introduced to pursue consistent representations between the structurally adjacent nodes and semantic neighbors. Despite their effectiveness, state-of-the-art contrastive recommender systems suffer from several inherent limitations: i) Graph augmentation with random perturbation may lose useful structural information, which misleads the representation learning. ii) The success of heuristic-guided representation contrasting schemes is largely built upon the view generator, which limits the model generality and is vulnerable to the noisy user behaviors. iii) Most of current GNN-based contrastive recommenders are limited by the over-smoothing issue which leads to indistinguishable representations.

In light of the above limitations and challenges, we revisit the graph contrastive learning paradigm for recommendation with a proposed simple yet effective augmentation method LightGCL. In our model, the graph augmentation is guided by singular value decomposition (SVD) to not only distill the useful information of user-item interactions but also inject the global collaborative context into the representation alignment of contrastive learning. Instead of generating two handcrafted augmented views, important semantic of user-item interactions can be well preserved with our robust graph contrastive learning paradigm. This enables our self-augmented representations to be reflective of both user-specific preferences and cross-user global dependencies.

Our contributions are highlighted as follows:

- In this paper, we enhance the recommender systems by designing a lightweight and robust graph contrastive learning framework to address the identified key challenges pertaining to this task.

- We propose an effective and efficient contrastive learning paradigm LightGCL for graph augmentation. With the injection of global collaborative relations, our model can mitigate the issues brought by inaccurate contrastive signals.

- Our method exhibits improved training efficiency compared to existing GCL-based approaches.

- Extensive experiments on several real-world datasets justify the performance superiority of our LightGCL. In-depth analyzes demonstrate the rationality and robustness of LightGCL.

## 2 RELATED WORK

**Graph Contrastive Learning for Recommendation**. A promising line of recent studies has incorporated contrastive learning (CL) into graph-based recommenders, to address the label sparsity issue with self-supervision signals. Particularly, SGL (Wu et al., 2021) and SimGCL (Yu et al., 2022a) perform data augmentation over graph structure and embeddings with random dropout operations. However, such stochastic augmentation may drop important information, which may make the sparsity issue of inactive users even worse. Furthermore, some recent alternative CL-based recommenders, such as HCCF (Xia et al., 2022b) and NCL (Lin et al., 2022), design heuristic-based strategies to construct view for embedding contrasting. Despite their effectiveness, their success heavily relies on their incorporated heuristics (e.g., the number of hyperedges or user clusters) for contrastive view generation, which can hardly be adaptive to different recommendation tasks.

**Self-Supervised Learning on Graphs**. Recently, self-supervised learning (SSL) has advanced the graph learning paradigm by enhancing node representation from unlabeled graph data (Zhu et al., 2021a;b; Velickovic et al., 2019; Hassani & Khasahmadi, 2020; Peng et al., 2020; Zhu et al., 2020; Wu et al., 2022). For example, to improve the predictive SSL paradigm, AutoSSL (Jin et al., 2022) automatically combines multiple pretext tasks for augmentation. Towards the line of contrastive SSL over graph structures, recent efforts focus on designing various graph contrastive learning methods (Yu et al., 2022b; Yin et al., 2022; Zhang et al., 2022; Xia et al., 2022a; Suresh et al., 2021). For instance, SimGRACE Xia et al. (2022a) proposes to generate contrastive views with the GNN encoder perturbations. In AutoGCL Yin et al. (2022), graph view generators are designed to be jointly trained with the graph encoder in an end-to-end way. Additionally, GCA (Zhu et al., 2021b) performs both topology-level and attribute-level data augmentation for contrastive view generation. In this method, important edges and features will be identified for adaptive augmentation. GraphCL (You et al., 2020) generates correlated graph representation views using various augmentation strategies, such as node/edge perturbation and attribute masking.

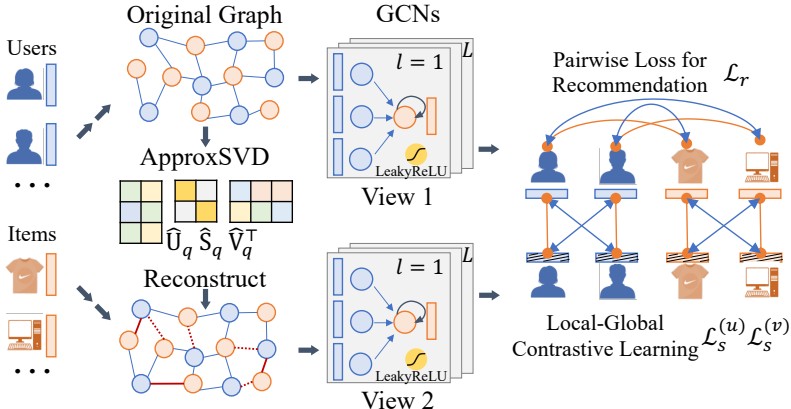

Figure 1: Overall structure of LightGCL.

## 3 METHODOLOGY

In this section, we describe our proposed LightGCL framework in detail. LightGCL is a lightweight graph contrastive learning paradigm as illustrated in Fig. 1. Complementary to the GCN backbone (the upper half of the figure) extracting the local graph dependency, the SVD-guided augmentation (the lower half of the figure) empowers the graph contrastive learning with global collaborative relation analysis for learning effective user and item representations.

### 3.1 LOCAL GRAPH DEPENDENCY MODELING

As a common practice of collaborative filtering, we assign each user $u_i$ and item $v_j$ with an embedding vector $\boldsymbol{e}_i^{(u)}, \boldsymbol{e}_j^{(v)} \in \mathbb{R}^d$, where $d$ is the embedding size. The collections of all user and item embeddings are defined as $\boldsymbol{E}^{(u)} \in \mathbb{R}^{I \times d}$ and $\boldsymbol{E}^{(v)} \in \mathbb{R}^{J \times d}$, where $I$ and $J$ are the number of users and items, respectively. Following Xia et al. (2022b), we adopt a two-layer GCN to aggregate the neighboring information for each node. In layer $l$, the aggregation process is expressed as follows:

$$\boldsymbol{z}_{i,l}^{(u)} = \sigma(p(\tilde{\mathcal{A}}_{i,:}) \cdot \boldsymbol{E}_{l-1}^{(v)}), \quad \boldsymbol{z}_{j,l}^{(v)} = \sigma(p(\tilde{\mathcal{A}}_{:,j}) \cdot \boldsymbol{E}_{l-1}^{(u)}) \tag{1}$$

where $\boldsymbol{z}_{i,l}^{(u)}$ and $\boldsymbol{z}_{j,l}^{(v)}$ denote the $l$-th layer aggregated embedding for user $u_i$ and item $v_j$. $\sigma(\cdot)$ represents the LeakyReLU with a negative slope of 0.5. $\tilde{\mathcal{A}}$ is the normalized adjacency matrix, on which we perform the edge dropout denoted as $p(\cdot)$, to mitigate the overfitting issue. We implement the residual connections in each layer to retain the original information of the nodes as follows:

$$\boldsymbol{e}_{i,l}^{(u)} = \boldsymbol{z}_{i,l}^{(u)} + \boldsymbol{e}_{i,l-1}^{(u)}, \quad \boldsymbol{e}_{j,l}^{(v)} = \boldsymbol{z}_{j,l}^{(v)} + \boldsymbol{e}_{j,l-1}^{(v)} \tag{2}$$

The final embedding for a node is the sum of its embeddings across all layers, and the inner product between the final embedding of a user $u_i$ and an item $v_j$ predicts $u_i$'s preference towards $v_j$:

$$\boldsymbol{e}_i^{(u)} = \sum_{l=0}^{L} \boldsymbol{e}_{i,l}^{(u)}, \quad \boldsymbol{e}_j^{(v)} = \sum_{l=0}^{L} \boldsymbol{e}_{j,l}^{(v)}, \quad \hat{y}_{i,j} = \boldsymbol{e}_i^{(u)\top} \boldsymbol{e}_j^{(v)} \tag{3}$$

### 3.2 EFFICIENT GLOBAL COLLABORATIVE RELATION LEARNING

To empower graph contrastive learning for recommendation with global structure learning, we equip our LightGCL with the SVD scheme (Rajwade et al., 2012; Rangarajan, 2001) to efficiently distill important collaborative signals from the global perspective. Specifically, we first perform SVD on the adjacency matrix $\mathcal{A}$ as $\mathcal{A} = \boldsymbol{U}\boldsymbol{S}\boldsymbol{V}^{\top}$. Here, $\boldsymbol{U}$ / $\boldsymbol{V}$ is an $I \times I$ / $J \times J$ orthonormal matrix with columns being the eigenvectors of $\mathcal{A}$'s row-row / column-column correlation matrix. $\boldsymbol{S}$ is an $I \times J$ diagonal matrix storing the singular values of $\mathcal{A}$. The largest singular values are usually associated

with the principal components of the matrix. Thus, we truncate the list of singular values to keep the largest q values, and reconstruct the adjacency matrix with the truncated matrices as $\hat{\mathcal{A}} = \boldsymbol{U}_q \boldsymbol{S}_q \boldsymbol{V}_q^\top$, where $\boldsymbol{U}_q \in \mathbb{R}^{I \times q}$ and $\boldsymbol{V}_q \in \mathbb{R}^{J \times q}$ contain the first $q$ columns of $\boldsymbol{U}$ and $\boldsymbol{V}$ respectively. $\boldsymbol{S}_q \in \mathbb{R}^{q \times q}$ is the diagonal matrix of the $q$ largest singular values.

The reconstructed matrix $\hat{\mathcal{A}}$ is a low-rank approximation of the adjacency matrix $\mathcal{A}$, for it holds that $rank(\hat{\mathcal{A}}) = q$. The advantages of SVD-based graph structure learning are two-folds. Firstly, it emphasizes the principal components of the graph by identifying the user-item interactions that are important and reliable to user preference representations. Secondly, the generated new graph structures preserve the global collaborative signals by considering each user-item pair. Given the $\hat{\mathcal{A}}$, we perform message propagation on the reconstructed user-item relation graph in each layer:

$$\boldsymbol{g}_{i,l}^{(u)} = \sigma(\hat{\mathcal{A}}_{i,:} \cdot \boldsymbol{E}_{l-1}^{(v)}), \quad \boldsymbol{g}_{j,l}^{(v)} = \sigma(\hat{\mathcal{A}}_{:,j} \cdot \boldsymbol{E}_{l-1}^{(u)}) \tag{4}$$

However, performing the exact SVD on large matrices is highly expensive, making it impractical for handling large-scale user-item matrix. Therefore, we adopt the randomized SVD algorithm proposed by Halko et al. (2011), whose key idea is to first approximate the range of the input matrix with a low-rank orthonormal matrix, and then perform SVD on this smaller matrix.

$$\hat{\boldsymbol{U}}_q, \hat{\boldsymbol{S}}_q, \hat{\boldsymbol{V}}_q^\top = \text{ApproxSVD}(\mathcal{A}, q), \quad \hat{\mathcal{A}}_{SVD} = \hat{\boldsymbol{U}}_q \hat{\boldsymbol{S}}_q \hat{\boldsymbol{V}}_q^\top \tag{5}$$

where $q$ is the required rank for the decomposed matrices, and $\hat{\boldsymbol{U}}_q \in \mathbb{R}^{I \times q}, \hat{\boldsymbol{S}}_q \in \mathbb{R}^{q \times q}, \hat{\boldsymbol{V}}_q \in \mathbb{R}^{J \times q}$ are the approximated versions of $\boldsymbol{U}_q, \boldsymbol{S}_q, \boldsymbol{V}_q$. Thus, we rewrite the message propagation rules in Eq. 4 with the approximated matrices and the collective representations of the embeddings as follows:

$$\boldsymbol{G}_l^{(u)} = \sigma(\hat{\mathcal{A}}_{SVD} \boldsymbol{E}_{l-1}^{(v)}) = \sigma(\hat{\boldsymbol{U}}_q \hat{\boldsymbol{S}}_q \hat{\boldsymbol{V}}_q^\top \boldsymbol{E}_{l-1}^{(v)}); \quad \boldsymbol{G}_l^{(v)} = \sigma(\hat{\mathcal{A}}_{SVD}^\top \boldsymbol{E}_{l-1}^{(u)}) = \sigma(\hat{\boldsymbol{V}}_q \hat{\boldsymbol{S}}_q \hat{\boldsymbol{U}}_q^\top \boldsymbol{E}_{l-1}^{(u)}) \tag{6}$$

where $\boldsymbol{G}_l^{(u)}$ and $\boldsymbol{G}_l^{(v)}$ are the collections of user and item embeddings encoded from the new generated graph structure view. Note that we do not need to compute and store the large dense matrix $\hat{\mathcal{A}}_{SVD}$. Instead, we can store $\hat{\boldsymbol{U}}_q$, $\hat{\boldsymbol{S}}_q$ and $\hat{\boldsymbol{V}}_q$, which are of low dimensions. By pre-calculating $(\hat{\boldsymbol{U}}_q \hat{\boldsymbol{S}}_q)$ and $(\hat{\boldsymbol{V}}_q \hat{\boldsymbol{S}}_q)$ during the preprocessing stage with SVD, the model efficiency is improved.

### 3.3 SIMPLIFIED LOCAL-GLOBAL CONTRASTIVE LEARNING

The conventional GCL methods such as SGL and SimGCL contrast node embeddings by constructing two extra views, while the embeddings generated from the original graph (the main-view) are not directly involved in the InfoNCE loss. The reason for adopting such a cumbersome three-view paradigm may be that the random perturbation used to augment the graph may provide misleading signals to the main-view embeddings. In our proposed method, however, the augmented graph view is created with global collaborative relations, which can enhance the main-view representations. Therefore, we simplify the CL framework by directly contrasting the SVD-augmented view embeddings $\boldsymbol{g}_{i,l}^{(u)}$ with the main-view embeddings $\boldsymbol{z}_{i,l}^{(u)}$ in the InfoNCE loss (Oord et al., 2018):

$$\mathcal{L}_s^{(u)} = \sum_{i=0}^{I} \sum_{l=0}^{L} -\log \frac{\exp(s(\boldsymbol{z}_{i,l}^{(u)}, \boldsymbol{g}_{i,l}^{(u)}/\tau))}{\sum_{i'=0}^{I} \exp(s(\boldsymbol{z}_{i,l}^{(u)}, \boldsymbol{g}_{i',l}^{(u)})/\tau)} \tag{7}$$

where $s(\cdot)$ and $\tau$ stand for the cosine similarity and the temperature respectively. The InfoNCE loss $\mathcal{L}_s^{(v)}$ for the items are defined in the same way. To prevent overfitting, we implement a random node dropout in each batch to exclude some nodes from participating in the contrastive learning. As shown in Eq. 8, the contrastive loss is jointly optimized with our main objective function for the recommendation task (where $\hat{y}_{i,p_s}$ and $\hat{y}_{i,n_s}$ denote the predicted scores for a pair of positive and negative items of user $i$):

$$\mathcal{L} = \mathcal{L}_r + \lambda_1 \cdot (\mathcal{L}_s^{(u)} + \mathcal{L}_s^{(v)}) + \lambda_2 \cdot \|\Theta\|_2^2; \quad \mathcal{L}_r = \sum_{i=0}^{I} \sum_{s=1}^{S} \max(0, 1 - \hat{y}_{i,p_s} + \hat{y}_{i,n_s}) \tag{8}$$

## 4 EVALUATION

To verify the superiority and effectiveness of the proposed LightGCL method, we perform extensive experiments to answer the following research questions:

- **RQ1**: How does LightGCL perform on different datasets compared to various SOTA baselines?
- **RQ2**: How does the lightweight graph contrastive learning improve the model efficiency?
- **RQ3**: How does our model perform against data sparsity, popularity bias and over-smoothing?
- **RQ4**: How does the local-global contrastive learning contribute to the performance of our model?
- **RQ5**: How do different parameter settings affect our model performance?

### 4.1 EXPERIMENTAL SETTINGS

#### 4.1.1 DATASETS AND EVALUATION PROTOCOLS

We evaluate our model and the baselines on five real-world datasets: **Yelp** (29,601 users, 24,734 items, 1,517,326 interactions): a dataset collected from the rating interactions on Yelp platform; **Gowalla** (50,821 users, 57,440 items, 1,172,425 interactions): a dataset containing users' check-in records collected from Gowalla platform; **ML-10M** (69,878 users, 10,195 items, 9,988,816 interactions): a well-known movie-rating dataset for collaborative filtering; **Amazon-book** (78,578 users, 77,801 items, 2,240,156 interactions): a dataset composed of users' ratings on books collected from Amazon; and **Tmall** (47,939 users, 41,390 items, 2,357,450 interactions): a E-commerce dataset containing users' purchase records on different products in Tmall platform.

In accordance with He et al. (2020) and Wu et al. (2021), we split the datasets into training, validation and testing sets with a ratio of 7:2:1. We adopt the Recall@N and Normalized Discounted Cumulative Gain (NDCG)@N, where N = $\{20, 40\}$, as the evaluation metrics.

#### 4.1.2 BASELINE METHODS

We compare our model against 16 state-of-the-art baselines with different learning paradigms:

- MLP-enhanced Collaborative Filtering: **NCF** (He et al., 2017).
- GNN-based Collaborative Filtering: **GCCF** (Chen et al., 2020c), **LightGCN** (He et al., 2020).
- Disentangled Graph Collaborative Filtering: **DGCF** (Wang et al., 2020b).
- Hypergraph-based Collaborative Filtering: **HyRec** (Wang et al., 2020a).
- Self-Supervised Learning Recommender Systems: **GraphCL** (You et al., 2020), **GRACE** (Zhu et al., 2020), **GCA** (Zhu et al., 2021b), **MHCN** (Yu et al., 2021), **SAIL** (Yu et al., 2022b), **AutoGCL** (Yin et al., 2022), **SimGRACE** (Xia et al., 2022a), **SGL** (Wu et al., 2021), **HCCF** (Xia et al., 2022b), **SHT** (Xia et al., 2022c), **SimGCL** (Yu et al., 2022a).

Due to space limit, the detailed descriptions of baselines are presented in Appendix A.

#### 4.1.3 HYPERPARAMETER SETTINGS

To ensure a fair comparison, we tune the hyperparameters of all the baselines within the ranges suggested in the original papers, except the following fixed settings for all the models: the embedding size is set as 32; the batch size is 256; two convolutional layers are used for GCN models.

For our LightGCL, the regularization weights $\lambda_1$ and $\lambda_2$ are tuned from $\{$1e-5, 1e-6, 1e-7$\}$ and $\{$1e-4, 1e-5$\}$, respectively. The temperature $\tau$ is searched from $\{0.3, 0.5, 1, 3, 10\}$. The dropout rate is chosen from $\{0, 0.25\}$. The rank (i.e., $q$) for SVD, is set as 5. We use the Adam optimizer with a learning rate of 0.001 decaying at the rate of 0.98 until the rate reaches 0.0005.[*]

### 4.2 PERFORMANCE VALIDATION (RQ1)

We summarize the experimental result in Table 1[†], with the following observations and conclusions:

---

[*]More detailed parameter settings can be found in our released source code.
[†]Due to space limit, results of NCF, GCCF, GraphCL, SAIL, GRACE, and AutoGCL are in Appendix B.

Table 1: Performance comparison with baselines on five datasets.

| Data | Metric | DGCF | HyRec | LightGCN | MHCN | SGL | SimGRACE | GCA | HCCF | SHT | SimGCL | LightGCL | *p-val.* | *impr.* |
|------|--------|------|-------|----------|------|-----|----------|-----|------|-----|--------|----------|----------|---------|
| Yelp | R@20 | 0.0466 | 0.0472 | 0.0482 | 0.0503 | 0.0526 | 0.0603 | 0.0621 | 0.0626 | 0.0651 | 0.0718 | **0.0793** | 7e-9 | 10% |
|      | N@20 | 0.0395 | 0.0395 | 0.0409 | 0.0424 | 0.0444 | 0.0435 | 0.0530 | 0.0527 | 0.0546 | 0.0615 | **0.0668** | 8e-9 | 8% |
|      | R@40 | 0.0774 | 0.0791 | 0.0803 | 0.0826 | 0.0869 | 0.0989 | 0.1021 | 0.1040 | 0.1091 | 0.1166 | **0.1292** | 2e-9 | 10% |
|      | N@40 | 0.0511 | 0.0522 | 0.0527 | 0.0544 | 0.0571 | 0.0656 | 0.0677 | 0.0681 | 0.0709 | 0.0778 | **0.0852** | 2e-9 | 9% |
| Gowalla | R@20 | 0.0944 | 0.0901 | 0.0985 | 0.0955 | 0.1030 | 0.0869 | 0.0896 | 0.1070 | 0.1232 | 0.1357 | **0.1578** | 1e-6 | 16% |
|      | N@20 | 0.0522 | 0.0498 | 0.0593 | 0.0574 | 0.0623 | 0.0528 | 0.0537 | 0.0644 | 0.0731 | 0.0818 | **0.0935** | 2e-6 | 14% |
|      | R@40 | 0.1401 | 0.1356 | 0.1431 | 0.1393 | 0.1500 | 0.1276 | 0.1322 | 0.1535 | 0.1804 | 0.1956 | **0.2245** | 3e-6 | 14% |
|      | N@40 | 0.0671 | 0.0660 | 0.0710 | 0.0689 | 0.0746 | 0.0637 | 0.0651 | 0.0767 | 0.0881 | 0.0975 | **0.1108** | 3e-6 | 13% |
| ML-10M | R@20 | 0.1763 | 0.1801 | 0.1789 | 0.1497 | 0.1833 | 0.2254 | 0.2145 | 0.2219 | 0.2173 | 0.2265 | **0.2613** | 1e-9 | 15% |
|      | N@20 | 0.2101 | 0.2178 | 0.2128 | 0.1814 | 0.2205 | 0.2686 | 0.2613 | 0.2629 | 0.2573 | 0.2613 | **0.3106** | 3e-9 | 18% |
|      | R@40 | 0.2681 | 0.2685 | 0.2650 | 0.2250 | 0.2768 | 0.3295 | 0.3231 | 0.3265 | 0.3211 | 0.3345 | **0.3799** | 7e-10 | 13% |
|      | N@40 | 0.2340 | 0.2340 | 0.2322 | 0.1962 | 0.2426 | 0.2939 | 0.2871 | 0.2880 | 0.3318 | 0.2880 | **0.3387** | 1e-9 | 17% |
| Amazon | R@20 | 0.0211 | 0.0302 | 0.0319 | 0.0296 | 0.0327 | 0.0381 | 0.0309 | 0.0322 | 0.0441 | 0.0474 | **0.0585** | 2e-7 | 23% |
|      | N@20 | 0.0154 | 0.0236 | 0.0236 | 0.0219 | 0.0249 | 0.0291 | 0.0238 | 0.0247 | 0.0328 | 0.0360 | **0.0436** | 2e-6 | 21% |
|      | R@40 | 0.0351 | 0.0432 | 0.0499 | 0.0489 | 0.0531 | 0.0621 | 0.0498 | 0.0525 | 0.0719 | 0.0750 | **0.0933** | 1e-7 | 24% |
|      | N@40 | 0.0201 | 0.0246 | 0.0290 | 0.0284 | 0.0312 | 0.0371 | 0.0301 | 0.0314 | 0.0420 | 0.0451 | **0.0551** | 9e-7 | 22% |
| Tmall | R@20 | 0.0235 | 0.0233 | 0.0225 | 0.0203 | 0.0268 | 0.0222 | 0.0373 | 0.0314 | 0.0387 | 0.0473 | **0.0528** | 3e-5 | 11% |
|      | N@20 | 0.0163 | 0.0160 | 0.0154 | 0.0139 | 0.0183 | 0.0152 | 0.0252 | 0.0213 | 0.0262 | 0.0328 | **0.0361** | 1e-4 | 10% |
|      | R@40 | 0.0394 | 0.0350 | 0.0378 | 0.0340 | 0.0446 | 0.0367 | 0.0616 | 0.0519 | 0.0645 | 0.0766 | **0.0852** | 1e-5 | 11% |
|      | N@40 | 0.0218 | 0.0199 | 0.0208 | 0.0188 | 0.0246 | 0.0203 | 0.0337 | 0.0284 | 0.0352 | 0.0429 | **0.0473** | 7e-5 | 10% |

- **Contrastive Learning Dominates.** As can be seen from the table, recent methods implementing contrastive learning (SGL, HCCF, SimGCL) exhibit consistent superiority as compared to traditional graph-based (GCCF, LightGCN) or hypergraph-based (HyRec) models. They also perform better than some of other self-supervised learning approaches (MHCN). This could be attributed to the effectiveness of CL to learn evenly distributed embeddings (Yu et al., 2022a).

- **Contrastive Learning Enhancement.** Our method consistently outperforms all the contrastive learning baselines. We attribute such performance improvement to the effective augmentation of graph contrastive learning via injecting global collaborative contextual signals. Other compared contrastive learning-based recommenders (e.g., SGL, SimGCL, and HCCF) are easily biased by noisy interaction information and generate misleading self-supervised signals.

## 4.3 EFFICIENCY STUDY (RQ2)

GCL models often suffer from a high computational cost due to the construction of extra views and the convolution operations performed on them during training. However, the low-rank nature of the SVD-reconstructed graph and the simplified CL structure enable the training of our LightGCL to be highly efficient. We analyze the pre-processing and per-batch training complexity of our model in comparison to three competitive baselines, as summarized in Table 2.[‡]

Table 2: Comparisons of computational complexity against baselines.

| Stage | Computation | LightGCN | SGL | SimGCL | LightGCL |
|-------|-------------|----------|-----|--------|----------|
| Pre-processing | Normalization | $O(E)$ | $O(E)$ | $O(E)$ | $O(E)$ |
|                | SVD | – | – | – | $O(qE)$ |
| Training | Augmentation | – | $O(2\rho E)$ | – | – |
|          | Graph Convolution | $O(2ELd)$ | $O(2ELd + 4\rho ELd)$ | $O(6ELd)$ | $O[2ELd + 2q(I+J)Ld]$ |
|          | BPR Loss | $O(2Bd)$ | $O(2Bd)$ | $O(2Bd)$ | $O(2Bd)$ |
|          | InfoNCE Loss | – | $O(Bd + BMd)$ | $O(Bd + BMd)$ | $O[(Bd + BMd)L]$ |

- Although our model requires performing the SVD in the pre-processing stage which takes $O(qE)$, the computational cost is negligible compared to the training stage since it only needs to be performed once. In fact, by moving the construction of contrastive view to the pre-processing stage, we avoid the repetitive graph augmentation during training, which improves model efficiency.

- Traditional GCN methods (e.g., LightGCN) only perform convolution on one graph, inducing a complexity of $O(2ELd)$ per batch. For most GCL-based methods, three contrastive views are

---

[‡]In the table, $E$, $L$ and $d$ denotes the edge number, the layer number and embedding size; $\rho \in (0, 1]$ is the edge keep rate; $q$ is the required rank; $I$ and $J$ represents the number of users and items; $B$ and $M$ are the batch size and node number in a batch. Detailed calculations are shown in Appendix D

computed per batch, leading to a complexity of roughly three times of LightGCN. In our model, instead, only two contrastive views are involved. Additionally, due to the low-rank property of SVD-based graph structure learning, our graph encoder takes only $O[2q(I + J)Ld]$ time. For most datasets, including the five we use, $2q(I + J) < E$. Therefore, the training complexity of our model is less than half of that of the SOTA efficient model SimGCL.

## 4.4 RESISTANCE AGAINST DATA SPARSITY AND POPULARITY BIAS (RQ3)

To evaluate the robustness of our model in alleviating data sparsity, we group the sparse users by their interaction degrees and calculate the *Recall@20* of each group on *Yelp* and *Gowalla* datasets. As can be seen from the figures, the performance of HCCF and SimGCL varies across datasets, but our LightGCL consistently outperforms them in all cases. In particular, our model performs notably well on the extremely sparse user group ($< 15$ interactions), as the *Recall@20* of these users is not much lower (and is even higher on *Gowalla*) than that of the whole dataset.

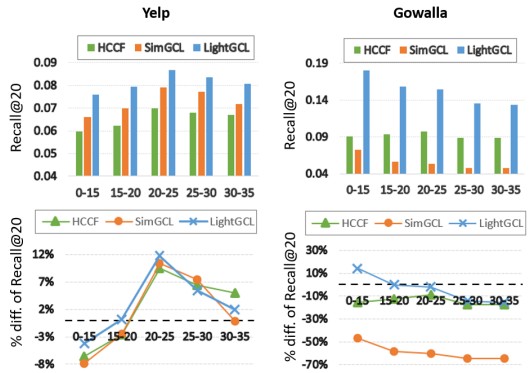

Figure 2: Performance on users of different sparsity degrees, in terms of *Recall* (histograms) and relative *Recall w.r.t* overall performances (charts).

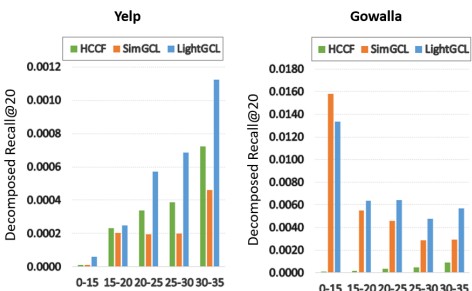

Figure 3: LightGCL's ability to alleviate popularity bias in comparison to SOTA CL-based methods HCCF and SimGCL.

Additionally, we illustrate our model's ability to mitigate popularity bias compared to HCCF and SimGCL. Similar to Section 4.4, we group the long-tail items by their degree of interactions. Following Wu et al. (2021), we adopt the decomposed *Recall@20* defined as $Recall^{(g)} = \frac{|(\mathbb{V}_{rec}^u)^{(g)} \cap \mathbb{V}_{test}^u|}{|\mathbb{V}_{test}^u|}$ where $\mathbb{V}_{test}^u$ refers to the set of test items for the user $u$, and $(\mathbb{V}_{rec}^u)^{(g)}$ is the set of Top-K recommended items for $u$ that belong to group $g$. The results are shown in Fig. 3. Similar to the results on sparse users, HCCF and SimGCL's performance fluctuates a lot with the influence of popularity bias. Our model performs better in most cases, which shows its resistance against popularity bias. Note that since the extremely sparse group ($< 15$ interactions) is significantly larger than the other groups in *Gowalla*, they contribute to a large fraction of the *Recall@20*, resulting in a different trend from that of *Yelp* in the figure.

## 4.5 BALANCING BETWEEN OVER-SMOOTHING AND OVER-UNIFORMITY (RQ3)

In this section, we illustrate the effectiveness of our model in learning a moderately dispersed embedding distribution, by preserving user unique preference pattern and inter-user collaborative dependencies. We randomly sample 2,000 nodes from *Yelp* and *Gowalla* and map their embeddings to the 2-D space with t-SNE (Van der Maaten & Hinton, 2008). The visualizations of these embeddings are presented in Fig. 4. We also calculate the Mean Average Distance (MAD) (Chen et al., 2020a) of the embeddings, summarized in Table 3.

Table 3: Mean Average Distance (MAD) of the embeddings learned by different methods.

| Dataset | MHCN | LightGCN | LightGCL | SGL | SimGCL |
|---------|------|----------|----------|-----|--------|
| Yelp | 0.8806 | 0.9469 | 0.9657 | 0.9962 | 0.9956 |
| Gowalla | 0.9247 | 0.9568 | 0.9721 | 0.9859 | 0.9897 |

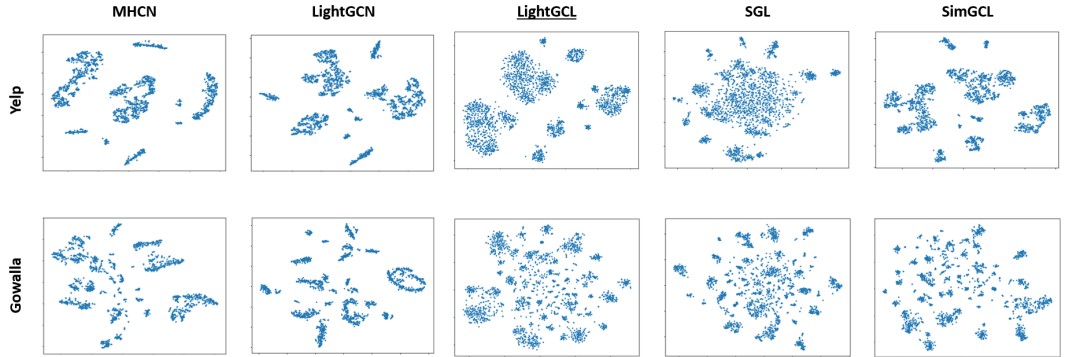

Figure 4: Embedding distributions on *Yelp* and *Gowalla* visualized with t-SNE.

As can be seen from Fig. 4, the embedding distributions of non-CL methods (i.e., LightGCN, MHCN) exhibit indistinguishable clusters in the embedding space, which indicates the limitation of addressing the over-smoothing issue. On the contrary, the existing CL-based methods tend to learn i) over-uniform distributions, e.g., SGL on *Yelp* learns a huge cloud of evenly-distanced embeddings with no clear community structure to well capture the collaborative relations between users; ii) highly dispersed small clusters with severe over-smoothing issue inside the clusters, e.g., the embeddings of SimGCL on *Gowalla* appear to be scattered grained clusters inside which embeddings are highly similar. Compared with them, clear community structures could be identified by our method to capture collaborative effects, while the embeddings inside each community are reasonably dispersed to be reflective of user-specific preference. The MAD of our model's learned features is also in between of the two types of baselines as shown in Table 3.

## 4.6 ABLATION STUDY (RQ4)

To investigate the effectiveness of our SVD-based graph augmentation scheme, we perform the ablation study to answer the question of whether we could provide guidance to the contrastive learning with a different approach of matrix decomposition. To this end, we implement two variants of our model, replacing the approximated SVD algorithm with other matrix decomposition methods: *CL-MF* adopts the view generated by a pre-trained MF (Koren et al., 2009); *CL-SVD++* utilizes the SVD++ (Koren, 2008) which takes implicit user feedback into consideration. As shown in Table 4, with the information distilled from MF or SVD++, the model is able to achieve satisfactory results, indicating the effectiveness of using matrix decomposition to empower CL and the flexibility of our proposed framework. However, adopting a pre-trained CL component is not only tedious and time-consuming but also inferior to utilizing the approximate SVD algorithm in terms of performance.

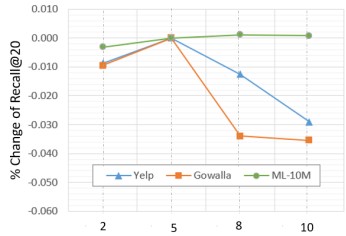

Figure 5: *Recall* change *w.r.t. q*.

Table 4: Ablation study on LightGCL.

| Variant | Yelp | | Gowalla | |
|---|---|---|---|---|
| | Recall@20 | NDCG@20 | Recall@20 | NDCG@20 |
| CL-MF | 0.0781 | 0.0659 | 0.1561 | 0.0929 |
| CL-SVD++ | 0.0788 | 0.0666 | 0.1568 | 0.0932 |
| LightGCL | **0.0793** | **0.0668** | **0.1578** | **0.0935** |

## 4.7 HYPERPARAMETER ANALYSIS (RQ5)

In this section, we investigate our model's sensitivity in relation to several key hyperparameters: the regularization weight for InfoNCE loss $\lambda_1$, the temperature $\tau$, and the required rank of SVD $q$.

- *The impact of $\lambda_1$.* As illustrated in Fig. 6, for the three datasets *Yelp*, *Gowalla* and *ML-10M*, the model's performance reaches the peak when $\lambda_1 = 10^{-7}$. It can be noticed that $\lambda_1$ with the range of $[10^{-6}, 10^{-8}]$ can often lead to performance improvement.

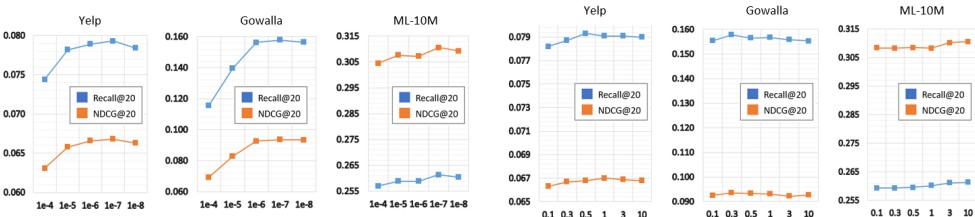

Figure 6: Impact of $\lambda_1$.        Figure 7: Impact of $\tau$

- *The impact of $\tau$.* Fig. 7 indicates that the model's performance is relatively stable across different selections of $\tau$ from 0.1 to 10, while the best configuration of $\tau$ value varies by datasets.

- *The selection of $q$.* $q$ determines the rank of SVD in our model. Experiments have shown that satisfactory results can be achieved with a small $q$. Specifically, as in Fig. 5, we observe that $q = 5$ is sufficient to preserve important structures of the user-item interaction graph.

## 4.8 CASE STUDY (RQ4)

In this section, we present a case study to intuitively show the effectiveness of our model to identify useful knowledge from noisy user-item interactions and make accurate recommendations accordingly. In Fig. 8, we can see that the venues visited by user #26 in *Yelp* mainly fall into two communities: Cleveland (where the user probably lives) and Arizona (where the user may have travelled to). In the reconstructed graph, these venues are assigned a new weight according to their potential importance. Note that item #2583, a car rental agency in Arizona, has been assigned a negative weight, which conforms to our common sense that people generally would not visit multiple car rental agencies in one trip. The SVD-augmented view also provides predictions on invisible links by assigning a large weight[§] to potential venues of interest, such as #2647 and #658. Note that when exploiting the graph, augmented view does not overlook the smaller Arizona community, which enables the model to predict items of minor interests that are usually overshadowed by the majority.

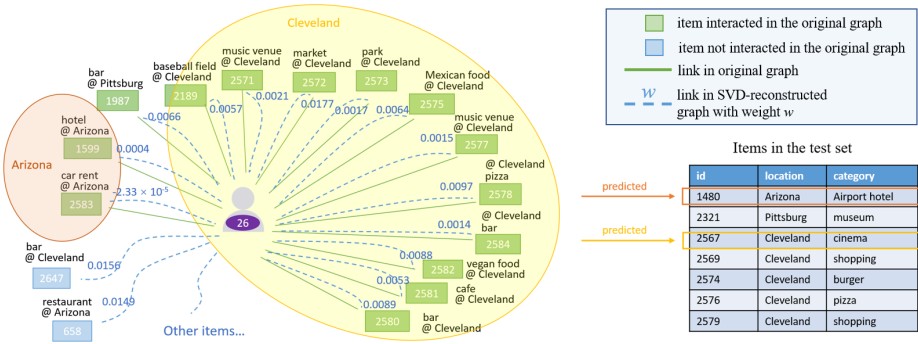

Figure 8: Case study on user #26 in *Yelp* dataset.

## 5 CONCLUSION

In this paper, we propose a simple and effective augmentation method to the graph contrastive learning framework for recommendation. Specifically, we explore the key idea of making the singular value decomposition powerful enough to augment user-item interaction graph structures. Our key findings indicate that our graph augmentation scheme exhibits strong ability in resisting data sparsity and popularity bias. Extensive experiments show that our model achieves new state-of-the-art results on several public evaluation datasets. In future work, we plan to explore the potential of incorporating casual analysis into our lightweight graph contrastive learning model to enhance the recommender system with mitigating confounding effects for data augmentation.

---

[§]Due to the fully connected nature of the SVD-reconstructed graph, the weights of unobserved interactions in the graph are of smaller magnitude. A weight of 0.01 is already a large weight in the graph.

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

## A    DETAILS OF THE BASELINES

MLP-enhanced Collaborative Filtering:

- **NCF** (He et al., 2017) is a collaborative filtering model that leverages neural network to exploit non-linearity. Two hidden layers are used in our evaluation.

GNN-based Collaborative Filtering:

- **GCCF** (Chen et al., 2020c) strengthens the GNN-based collaborative filtering by implementing a residual network and reducing the non-linear transformation.
- **LightGCN** (He et al., 2020) adopts a simplified GCN structure without embedding weight matrices and non-linear projection.

Disentangled Graph Collaborative Filtering:

- **DGCF** (Wang et al., 2020b) learns a more sophisticated representation by segmenting the embedding vectors to represent multiple latent intentions.

Hypergraph-based Collaborative Filtering:

- **HyRec** (Wang et al., 2020a) makes use of hypergraph to encode multi-order information between users and items.

Self-Supervised Learning Recommender Systems:

- **GraphCL** (You et al., 2020) utilizes random node dropping and edge masking to generate two contrastive views, which were aligned by optimizing the SSL loss function.
- **GRACE** (Zhu et al., 2020) proposes to corrupt the graph structure by both random edge dropout and random node feature dropping, and uses the corrupted graphs as the contrastive views.
- **GCA** (Zhu et al., 2021b) adaptively dropout the nodes and edges by their importance calculated with node centrality.
- **MHCN** (Yu et al., 2021) creates self-supervised signals for the graph representation learning by graph infomax network.
- **SAIL** (Yu et al., 2022b) maximizes the neighborhood predicting probability between GNN-generated high-level features and input node features.
- **AutoGCL** (Yin et al., 2022) uses GNN to learn to mask nodes and edges in the augmented graph. It minimizes the similarity between the augmented and the original graph, while maximizing the similarity of the embeddings generated through them, so as to uncover the most important information in the graph.
- **SimGRACE** (Xia et al., 2022a) creates augmented view by randomly perturbing the parameters of the GNN network.
- **SGL** (Wu et al., 2021) adopts random walk sampling and probabilistic edge/node dropout to create augmented views for contrastive learning. In our experiments, we adopt the SGL-ED variant, which implements random edge dropout and exhibits the strongest performance according to the original paper.
- **HCCF** (Xia et al., 2022b) encodes global graph information with hypergraph and contrasts it against the local information encoded with GCN. In our experiments, the number of hyper-edges are set as 128 following the original paper.
- **SHT** (Xia et al., 2022c) adopts a hypergraph transformer framework to exploit global collaborative relationships and distills the global information to generate the cross-view self-supervised signals. In our experiments, the number of hyper-edges are set as 128 following the original paper.
- **SimGCL** (Yu et al., 2022a) propose to simplify the graph augmentation process of contrastive learning by directly injecting random noises into the feature representation.

## B  PERFORMANCE COMPARISON WITH BASELINES (CONTINUED)

In this appendix, we show the performance of NCF, GCCF, GraphCL, SAIL, GRACE, and Auto-GCL, which are not shown in Table 1 due to space limit. The results are summarized in Table 5. As can be seen from the table, our model outperforms these baselines consistently.

Table 5: Performance comparison with baselines on five datasets (continued).

| Data | Metric | NCF | GCCF | GraphCL | SAIL | GRACE | AutoGCL | LightGCL |
|------|--------|------|------|---------|------|-------|---------|----------|
| Yelp | R@20 | 0.0252 | 0.0462 | 0.0462 | 0.0471 | 0.0550 | 0.0593 | **0.0793** |
|      | N@20 | 0.0202 | 0.0398 | 0.0401 | 0.0405 | 0.0470 | 0.0494 | **0.0668** |
|      | R@40 | 0.0487 | 0.0760 | 0.0764 | 0.0773 | 0.0917 | 0.1009 | **0.1292** |
|      | N@40 | 0.0289 | 0.0508 | 0.0511 | 0.0516 | 0.0605 | 0.0650 | **0.0852** |
| Gowalla | R@20 | 0.0171 | 0.0951 | 0.0997 | 0.0999 | 0.0744 | 0.0832 | **0.1578** |
|      | N@20 | 0.0106 | 0.0535 | 0.0603 | 0.0602 | 0.0452 | 0.0484 | **0.0935** |
|      | R@40 | 0.0216 | 0.1392 | 0.1473 | 0.1472 | 0.1071 | 0.1291 | **0.2245** |
|      | N@40 | 0.0118 | 0.0684 | 0.0727 | 0.0725 | 0.0539 | 0.0605 | **0.1108** |
| ML-10M | R@20 | 0.1097 | 0.1742 | 0.1659 | 0.1728 | 0.2107 | 0.2325 | **0.2613** |
|      | N@20 | 0.1297 | 0.2109 | 0.2038 | 0.2118 | 0.2476 | 0.2755 | **0.3106** |
|      | R@40 | 0.1634 | 0.2606 | 0.2560 | 0.2639 | 0.3075 | 0.3415 | **0.3799** |
|      | N@40 | 0.1427 | 0.2331 | 0.2250 | 0.2332 | 0.2711 | 0.3023 | **0.3387** |
| Amazon | R@20 | 0.0142 | 0.0317 | 0.0360 | 0.0357 | 0.0360 | 0.0325 | **0.0585** |
|      | N@20 | 0.0085 | 0.0243 | 0.0266 | 0.0264 | 0.0271 | 0.0241 | **0.0436** |
|      | R@40 | 0.0223 | 0.0483 | 0.0585 | 0.0581 | 0.0583 | 0.0553 | **0.0933** |
|      | N@40 | 0.0133 | 0.0285 | 0.0340 | 0.0338 | 0.0345 | 0.0318 | **0.0551** |
| Tmall | R@20 | 0.0082 | 0.0209 | 0.0251 | 0.0254 | 0.0303 | 0.0312 | **0.0528** |
|      | N@20 | 0.0059 | 0.0141 | 0.0175 | 0.0177 | 0.0210 | 0.0204 | **0.0361** |
|      | R@40 | 0.0140 | 0.0356 | 0.0416 | 0.0424 | 0.0505 | 0.0524 | **0.0852** |
|      | N@40 | 0.0079 | 0.0196 | 0.0233 | 0.0236 | 0.0281 | 0.0278 | **0.0473** |

## C  THEORETICAL ANALYSIS

We conduct theoretical analyses to show that our local-global CL (Eq. 7) is augmented to maximize the similarity between embeddings of potentially related nodes, based on the SVD-based global relation learning. Specifically, for a node $v_j \in \mathcal{U}$, where $\mathcal{U} = \{u_{i'}|\mathcal{A}_{i,i'} = 0, \hat{\mathcal{A}}_{i,i'} \neq 0\}$, the embeddings are not updated by $s(\boldsymbol{z}_{i,l}, \boldsymbol{g}_{i,l})$ in the vanilla InfoNCE loss, as $v_j$ is not adjacent to $u_i$. Instead, our local-global contrastive assigns the following gradients to the embeddings of $v_j$:

$$\partial s(\boldsymbol{z}_{i,l}, \boldsymbol{g}_{i,l})/\partial \boldsymbol{g}_{i,l-1} = \partial s\left(\boldsymbol{z}_{i,l}, \sigma(\sum_{j\in\mathcal{U}} \alpha_{i,j}\boldsymbol{g}_{j,l-1} + \sum_{\mathcal{A}_{i,j'}\neq 0} \alpha_{i,j'}\boldsymbol{g}_{j',l-1})\right)/\partial \boldsymbol{g}_{j,l-1}$$

$$= \frac{\boldsymbol{z}_{i,l}}{\|\boldsymbol{z}_{i,l}\|\|\boldsymbol{g}_{i,l}\|} \cdot \sigma'(\cdot) \cdot \alpha_{i,j} \tag{9}$$

where $\alpha_{i,j}$ denotes the normalization weight for node $u_i$ and $v_j$. In this way, the embeddings of nodes in $\mathcal{U}$ are also pulled close to $\boldsymbol{s}_{i,l}$, which injects relatedness information learned by the SVD into the local-global CL optimization.

## D  CALCULATION OF COMPLEXITY

### D.1  ADJACENCY MATRIX NORMALIZATION

For a sparse user-item matrix stored in the Coordinate Format (COO), it requires visiting every non-zero elements in the matrix to perform normalization. Thus, the computational complexity is in the order of the number of edges $O(E)$. Note that for the baseline SGL, it requires normalizing the two augmented graph structures during the training phase, each of which contains $\rho E$ edges, so it induces a complexity of $O(2\rho E)$ per batch.

### D.2  APPROXIMATE SVD ALGORITHM

We refer the readers to Halko et al. (2011) in which the complexity of the approximate SVD algorithm is explained in detail.

### D.3 GRAPH CONVOLUTION

Given a sparse COO matrix $\mathcal{A}$ with $E$ edges and a dense matrix $\boldsymbol{E}$ with dimensions $I(J) \times d$, it takes $O(Ed)$ time to calculate $\mathcal{A}\boldsymbol{E}$. To perform graph convolution on a graph, we need to multiply the sparse adjacency matrix with $\boldsymbol{E}_{l-1}^{(v)} \in \mathbb{R}^{J \times d}$ and its transpose with $\boldsymbol{E}_{l-1}^{(u)} \in \mathbb{R}^{I \times d}$, which takes $O(Ed)$ each, and $O(2Ed)$ in total. For $L$ layers, $O(2ELd)$ is required. For traditional CL-based methods such as SGL and SimCGL, a three-view structure is adopted, resulting in a complexity of $O(12ELd)$ (for SGL it again varies a bit depending on $\rho$).

For the SVD-view of our model, $\hat{\boldsymbol{V}}_q^\top \boldsymbol{E}_{l-1}^{(v)}$ takes $O(qJd)$, and multiplying the result with the pre-calculated $(\hat{\boldsymbol{U}}_q \hat{\boldsymbol{S}}_q)$ takes $O(qId)$; $\hat{\boldsymbol{U}}_q^\top \boldsymbol{E}_{l-1}^{(v)}$ takes $O(qId)$, and multiplying the result with the pre-calculated $(\hat{\boldsymbol{V}}_q \hat{\boldsymbol{S}}_q)$ takes $O(qJd)$. So in total it takes $O(2q(I+J)d)$.

### D.4 BPR LOSS

In each batch with $B$ users, calculating the scores for positive and negative items both take $O(Bd)$, so in total it takes $O(2Bd)$.

### D.5 CL LOSS

In each batch with $B$ users, calculating the numerator of InfoNCE loss takes $O(Bd)$, and calculating the denominator takes $O(BMd)$ where $M$ denotes the total number of nodes in the batch. Since our model adopts a per layer InfoNCE loss, a factor of $L$ is appended.

