# OpenReview forum: "LightGCL: Simple Yet Effective Graph Contrastive Learning for Recommendation"
_ICLR.cc/2023/Conference — ICLR 2023 notable top 25%_

### Official Review · Reviewer_k7CA · 2022-10-23

**Confidence:** 4
**Correctness:** 2
**Technical Novelty And Significance:** 2
**Empirical Novelty And Significance:** 2
**Recommendation:** 5

**Clarity, Quality, Novelty And Reproducibility:**

Compared with the existing work (Robust Tensor Graph Convolutional Networks via T-SVD based Graph Augmentation. KDD 2022), the highlights should be more prominent.

**Strength And Weaknesses:**

Strength:
(1) The idea of using SVD to to augment user-item interaction graph structures seems interesting to me.
(2)  The codes and datasets of this work are provided for reproducibility.
(3) The experimental results look promising and show relative improvements over baselines.

Weakness:
(1) The truncated T-SVD to capture the low-rankness of the multi-view augmented graph, which improves the robustness from the perspective of graph preprocessing in “Zhebin Wu, Lin Shu, Ziyue Xu, Yaomin Chang, Chuan Chen, Zibin Zheng: Robust Tensor Graph Convolutional Networks via T-SVD based Graph Augmentation. KDD 2022: 2090-2099”. Compared to this work, which also uses SVD-based graph augmentation and improved robustness, what are your highlights?
(2) This paper focuses on a general SVD-based graph structure learning, more variants based on SVD (eg. RSVD and SVD++, ) should be designed in the ablation study, instead of only a pre-trained MF to replace it.
(3) In Section 4.2, the authors did not provide statistical significance tests. The improvement ratio of the proposed LightGCL over the optimal baseline is also not stated.
(4) Figure 1 shows the overall structure of LightGCL, however, lack of overall description for this figure.
(5) As we all know, the disadvantage of SVD is that the decomposed matrix often lacks interpretation. Has the author considered this problem in recommendation?

Some papers involving graph contrastive learning are missed, such as:
-  Lu Yu, Shichao Pei, Lizhong Ding, Jun Zhou, Longfei Li, Chuxu Zhang, Xiangliang Zhang: SAIL: Self-Augmented Graph Contrastive Learning. AAAI 2022: 8927-8935.
-  Yihang Yin, Qingzhong Wang, Siyu Huang, Haoyi Xiong, Xiang Zhang: AutoGCL: Automated Graph Contrastive Learning via Learnable View Generators. AAAI 2022: 8892-8900.
- Yixin Zhang, Yong Liu, Yonghui Xu, Hao Xiong, Chenyi Lei, Wei He, Lizhen Cui, Chunyan Miao: Enhancing Sequential Recommendation with Graph Contrastive Learning. IJCAI 2022: 2398-2405.
- Jun Xia, Lirong Wu, Jintao Chen, Bozhen Hu, Stan Z. Li: SimGRACE: A Simple Framework for Graph Contrastive Learning without Data Augmentation. WWW 2022: 1070-1079.
- Susheel Suresh, Pan Li, Cong Hao, Jennifer Neville: Adversarial Graph Augmentation to Improve Graph Contrastive Learning. NeurIPS 2021: 15920-15933.

There are some typos to check, such as:
- “masked featurte reconstruction”—>“masked feature reconstruction” (page 2)
-“Normalized Discounted Culmulative Gain”—>“ Normalize Discounted Cumulative Gain” (page 5)
-“Addtionally”—>“ Additionally” (page 7)
-“while the best configuration of τ value vary by datasets”—>“ while the best configuration of τ value varies by datasets” (page 9)
-“we presents a case study”—>“ we presents a case study” (page 9)
-“our graph augmentation scheme exhibit strong ability ”—>“our graph augmentation scheme exhibits strong ability” (page 9)

**Summary Of The Paper:**

This paper presents a graph contrastive learning paradigm named LightGCL, the singular value decomposition is adopted for contrastive augmentation to achieve the unconstrained structure refinement with global collaborative relation modeling. Experiments on five datasets validate the effectiveness of the proposed LightGCL compared with ten baselines.

**Summary Of The Review:**

Overall, the idea of using a  SVD-based graph structure learning to achieve graph augmentation in contrastive learning is interesting and efficient. However, compared with the existing work (Robust Tensor Graph Convolutional Networks via T-SVD based Graph Augmentation. KDD 2022), the highlights should be more prominent. Moreover, some important relevant literatures is missing, and more SVD-based ablation studies should be discussed.

---

> ### Author Response · Authors · 2022-11-18
> **Response to Reviewer k7CA (1/3)**
>
> We really appreciate your detailed and thorough feedback on our submission! We have made modifications according to your suggestions. In response to your comments, further clarifications and experiments are added below with details. We really appreciate your kind consideration if you find our response helpful in addressing your comments.
>
> ### **1) Difference with RTGCN in the paper by Wu et al. in KDD'22.**
>
> **Response**: Thanks a lot for bringing this up. Our work has advantages over the RTGCN proposed in Wu's paper in the following four aspects. To address the reviewer's comment, we further cite this paper in the second paragraph of our related work section as relevant work.
>
> **i)** The augmentation method in RTGCN is still based on random perturbation, which may inject unwanted noises into the model. Specifcially, RTGCN first creates multiple augmented views from the original graph by randomly replacing some edges, and then perform T-SVD and truncation to build the low-rank approximations of the original graph. In this process, the randomly replaced edges may contain misleading information. However, in our model, the randomness is minimized and only exists in the randomized SVD algorithm. This makes the augmented view created by our model more meaningful.
>
> **ii)** In RTGCN, the original graph does not directly participate in the graph convolution. Instead, all the views in the tensor are augmented views by random perturbation. This may lead to a loss in essential information in the original graph. However, in our model, the original graph is directly involved in the training.
>
> **iii)** Our LightGCL adopts the contrastive learning paradigm, which can maximize the alignment between the views. In contrast, the multi-view augmentation approach adopted by RTGCN does not try to align different views, so it cannot enjoy the benefits brought by the contrastive learning-enhanced augmentation.
>
> **iv)** The most practical issue with RTGCN is that it is very difficult to apply it in our recommandation scenario because of the scale of our adopted datasets. The T-SVD is too computationally expensive to be run on major recommendation datasets. The high computational costs mainly come from two sources: (a) Tensor product is much more expensive than matrix product; (b) T-SVD uses exact SVD + truncation to do the low-rank approximation, which, as stated in our paper, is not possible to be performed on large datasets even for 2D matrices, let alone 3D tensors. In the paper of RTGCN, the datasets used are all very small, with several thousand edges per dataset, but in the datasets for recommendation, the number of edges is several millions. In fact, we have tried to compare RTGCN with our model on our datasets, but our device cannot run it. (Our device has GPU [NVIDIA GeForce RTX 3090](https://www.nvidia.com/en-us/geforce/graphics-cards/30-series/rtx-3090-3090ti/) and CPU [Intel® Xeon® Silver 4314](https://www.intel.com/content/www/us/en/products/sku/215269/intel-xeon-silver-4314-processor-24m-cache-2-40-ghz/specifications.html).)
>
> ### **2) More variants in the ablation study.**
>
> **Response**: Thanks for this valuable suggestion! In fact, the framework we propose is a flexible one, under which many SVD-like methods exploiting the graph information could demonstrate a plus to the model performance. We agree that more variants of our model should be tested to illustrate this point. Regarding the two methods you mentioned (RSVD and SVD++):
>
> **i)** Actually, the approximate SVD algorithm [1] we used in our paper is itself an RSVD method, because it is a randomized SVD algorithm that first finds the SVD of a smaller matrix associated with the original one and then uses the result to approximate the SVD of the original one.
>
> **ii)** We have tested the SVD++ [2] variant as you suggested. And the results have been added to the updated version of our paper. The performance of this variant lies between the CL-MF variant and LightGCL. However, it is worth noting that, similar to the pre-trained MF, the SVD++ is solved with gradient descent, so it requires pre-training to obtain the reconstructed adjacency matrix. Adopting such a pre-training method will increase the computational costs and reduce the model efficiency. Therefore, the best choice of the CL component is still the approximate SVD algorithm [1] adopted in our model.
>
> ### **3) Significance test and improvement ratio.**
>
> **Response**: Thanks for this suggestion. To address the reviewer's comment, we have added the *p-values* and the *improvement ratios* into Table 1 in the updated version of our paper. The *p-values* and *improvement ratios* demonstrated that our improvement over the strongest baseline (SimGCL) is statistically significant.

---

> > ### Author Response · Authors · 2022-11-18
> > **Response to Reviewer k7CA (2/3)**
> >
> > ### **4) Overall description of the model architecture figure.**
> >
> > **Response**: Thanks for pointing it out. We have added a short description of Fig. 1 to the first paragraph of the *Methodology* section in the updated version of our paper as follows:
> >
> > > In this section, we describe our proposed LightGCL framework in detail. LightGCL is a lightweight graph contrastive learning paradigm as illustrated in Fig. 1. Complementary to the GCN backbone (the upper half of the figure) extracting the local graph dependency, the SVD-guided augmentation (the lower half of the figure) empowers the graph contrastive learning with global collaborative relation analysis for learning effective user and item representations.
> >
> > ### **5) Interpretation of SVD.**
> >
> > **Response**: Thanks for this constructive feedback. As the reviewer suggested, the SVD method, which focuses on low-rank decomposition, may lack sufficient interpretation. In general, the matrix structures refined by our SVD component may filter out some redundant information and preserve important user-item interaction patterns. In our LightGCL, we leverage SVD-based decomposition to generate contrastive view over user-item interaction graph structures for efficient augmentation. The effectiveness and efficiency have been demonstrated through our empirical studies. We agree with the reviewer that the interpretation of SVD is an interesting direction to explore. We leave it as our future work for interpretable graph contrastive learning for recommender systems. One potential solution is to integrate graph information bottleneck with the SVD-based component, to enable the preserved important graph structures to be interpretable. To evaluate the model explanability, fidelity metric will be adopted. Note that high fidelity score indicates better model interpretability.

---

> > > ### Author Response · Authors · 2022-11-18
> > > **Response to Reviewer k7CA (3/3)**
> > >
> > > ### **6) Discussion and comparison with the suggested related work on GCL.**
> > >
> > > **Response**: Thank you for pointing it out. Based on the suggested relevant research works, we have updated our paper as follows:
> > >
> > > **i)** We added all the suggested relevant papers in our related work for comprehensive related work study.
> > >
> > > > Towards the line of contrastive SSL over graph structures, recent efforts focus on designing various graph contrastive learning methods (Yu et al., 2022c; Yin et al., 2022b; Zhang et al., 2022; Xia et al., 2022b; Suresh et al., 2021). For instance, SimGRACE Xia et al. (2022b) proposes to generate contrastive views with the GNN encoder perturbations. In AutoGCL Yin et al. (2022b), graph view generators are designed to be jointly trained with the graph encoder in an end-to-end way.
> > >
> > > **ii)** We included three of them into our performance comparisons: SimGRACE, SAIL, and AutoGCL. Particularly, SimGRACE is added to the main comparison table (Table 1). Due to space limit, we put the other two together with some other models into Table 5 in Appendix B.
> > >
> > > **iii)** In addition, we included three other GCL baselines into our comparisons: GraphCL, GRACE, and GCA.
> > >
> > > **iv)** Our model consistently outperformed the newly added baselines. Please refer to Table 1 and Table 5 in our updated paper for the details. We also attach the results of the newly added baselines here for your easy reference.
> > >
> > >  | Data    | Metric | GraphCL | SAIL   | GRACE  | AutoGCL | SimGRACE | GCA    | **LightGCL** |
> > > |---------|--------|---------|--------|--------|---------|----------|--------|--------------|
> > > | Yelp    | R@20   | 0.0462  | 0.0471 | 0.055  | 0.0593  | 0.0603   | 0.0621 | **0.0793**   |
> > > |         | N@20   | 0.0401  | 0.0405 | 0.047  | 0.0494  | 0.0435   | 0.053  | **0.0668**   |
> > > |         | R@40   | 0.0764  | 0.0773 | 0.0917 | 0.1009  | 0.0989   | 0.1021 | **0.1292**   |
> > > |         | N@40   | 0.0511  | 0.0516 | 0.0605 | 0.065   | 0.0656   | 0.0677 | **0.0852**   |
> > > | Gowalla | R@20   | 0.0997  | 0.0999 | 0.0744 | 0.0832  | 0.0869   | 0.0896 | **0.1578**   |
> > > |         | N@20   | 0.0603  | 0.0602 | 0.0452 | 0.0484  | 0.0528   | 0.0537 | **0.0935**   |
> > > |         | R@40   | 0.1473  | 0.1472 | 0.1071 | 0.1291  | 0.1276   | 0.1322 | **0.2245**   |
> > > |         | N@40   | 0.0727  | 0.0725 | 0.0539 | 0.0605  | 0.0637   | 0.0651 | **0.1108**   |
> > > | ML-10M  | R@20   | 0.1659  | 0.1728 | 0.2107 | 0.2325  | 0.2254   | 0.2145 | **0.2613**   |
> > > |         | N@20   | 0.2038  | 0.2118 | 0.2476 | 0.2755  | 0.2686   | 0.2613 | **0.3106**   |
> > > |         | R@40   | 0.2560  | 0.2639 | 0.3075 | 0.3415  | 0.3295   | 0.3231 | **0.3799**   |
> > > |         | N@40   | 0.2250  | 0.2332 | 0.2711 | 0.3023  | 0.2939   | 0.2871 | **0.3387**   |
> > > | Amazon  | R@20   | 0.0360  | 0.0357 | 0.036  | 0.0325  | 0.0381   | 0.0309 | **0.0585**   |
> > > |         | N@20   | 0.0266  | 0.0264 | 0.0271 | 0.0241  | 0.0291   | 0.0238 | **0.0436**   |
> > > |         | R@40   | 0.0585  | 0.0581 | 0.0583 | 0.0553  | 0.0621   | 0.0498 | **0.0933**   |
> > > |         | N@40   | 0.0340  | 0.0338 | 0.0345 | 0.0318  | 0.0371   | 0.0301 | **0.0551**   |
> > > | Tmall   | R@20   | 0.0251  | 0.0254 | 0.0303 | 0.0312  | 0.0222   | 0.0373 | **0.0528**   |
> > > |         | N@20   | 0.0175  | 0.0177 | 0.021  | 0.0204  | 0.0152   | 0.0252 | **0.0361**   |
> > > |         | R@40   | 0.0416  | 0.0424 | 0.0505 | 0.0524  | 0.0367   | 0.0616 | **0.0852**   |
> > > |         | N@40   | 0.0233  | 0.0236 | 0.0281 | 0.0278  | 0.0203   | 0.0337 | **0.0473**   |
> > >
> > >
> > >
> > > ### **7) Fixing typos.**
> > >
> > > **Response**: Sorry for the typos. We have corrected them in the updated version.
> > >
> > > **References**
> > >
> > >  [1] Nathan Halko, Per-Gunnar Martinsson, and Joel A Tropp. Finding structure with randomness: Probabilistic algorithms for constructing approximate matrix decompositions. *SIAM review*, 53(2):217–288, 2011.
> > >
> > > [2] Yehuda Koren. Factorization meets the neighborhood: a multifaceted collaborative filtering model. In *Proceedings of the 14th ACM SIGKDD international conference on Knowledge discovery and data mining*, pp. 426–434, 2008.

---

### Official Review · Reviewer_rCJG · 2022-10-24

**Confidence:** 5
**Correctness:** 4
**Technical Novelty And Significance:** 3
**Empirical Novelty And Significance:** 4
**Recommendation:** 8

**Clarity, Quality, Novelty And Reproducibility:**

The major novelty of enhancing graph contrastive learning by preserving important structural information is reasonable and effective. This paper is well-written and easy to follow. The presentation of methodology is very clear. The authors have released the code and models.

**Details Of Ethics Concerns:**

N.A.

**Strength And Weaknesses:**

Strong points:
1.The proposed method is well-motivated and technically sound, by addressing key challenges of current GCL-based recommender systems to automatically generate augmented graph view for effectively incorporating self-supervised learning signals.
2. A lightweight graph augmentation model is built over an efficiency SVD-based graph structure learning framework.
3.Comprehensive experiments show the superiority of the proposed method over SOTA baselines.
4. In-depth model evaluation demonstrates the advantage of the LightGCL approach for alleviating data sparsity, over-smoothing issue, and popularity bias.

Weak points:
1. For the compared SGL baseline method, three different augmentation schemes are introduced, e.g., node/edge dropout operators, as well as random walk-based sampling approach. The details of configuring the augmentation operators in SGL method can be further described.
2. For the proposed approach, its advantage lies in the automated graph augmentation via distilling important graph structural information using a decomposition-based method. Compared with baseline SHT, the advantage of LightGCL can be further clarified.
3. The results need further explanation. For instance, why the methods exhibit different trends on Yelp and Gowalla (Fig. 3).

**Summary Of The Paper:**

This work proposes a novel method for advancing recommender system with a lightweight and effective graph contrastive learning paradigm. The proposed method is technically sound from a novel perspective of efficient SVD-guided graph augmentation. To demonstrate the effectiveness of the new framework, 10 baselines are compared.

**Summary Of The Review:**

This paper tackles important challenges of graph contrastive learning in recommender systems. I acknowledge the technical novelty and empirical evidence of this work.

---

> ### Author Response · Authors · 2022-11-18
> **Response to Reviewer rCJG**
>
> Thank you so much for providing valuable suggestions and acknowledgement of our technical novelty and empirical studies. We make the following clarifications in response to your feedback.
>
> ### **1) The configuration of the baseline SGL (ND/ED/RW).**
>
> **Response**: Thanks for pointing it out. In our experiments, we used the SGL-ED variant which implements the random edge dropout, because this variant exhibits the strongest performance as shown in the original paper. We have added this description of configuration to the description of SGL in Appendix A on Page 12.
>
> ### **2) Further clarifying the advantage compared to SHT.**
>
> **Response**: In the baseline method SHT, an auxiliary learning task of user-item edge predictions is incorporated into the recommendation task for generative self-supervised learning based on hypergraph transformer architecture. There exist two limitations in SHT: i) the incorporated generative learning task heavily relies on the observed user-item interaction data, which can be misled by redundant and noisy information; ii) the adapted hypergraph transformer architecture is not computationally-efficient, due to the high cost of self-attention calculations for modeling node-hyperedge relationships. In contrast, our LightGCL not only injects the global collaborative signals for preserving important user-item interaction graph structures, but also enables highly efficient graph contrastive learning paradigm for recommendation because of the low-rank nature of singular value decomposition.
>
> ### **3) Explanations of the different trends on Yelp and Gowalla in Fig.3.**
>
> **Response**: Thanks for pointing this out. In fact, this is due to the different distributions of the two datasets. As can be seen from the following table describing the sizes of each group, the sizes of extremely sparse groups in Gowalla are significantly larger than the other groups, and thus they contribute to a large fraction of the Recall@20. In contrast, the distribution of Yelp is more even, so it follows the common trend in which performance is lower in sparser groups. We have added an explanation of it to the end of Section 4.4 in our revision.
>
> | | <=15 | 15-20 | 20-25 | 25-30 | 30-35 | >35 | Total |
> |--|--|--|--|--|--|--|--|
> | Yelp | 4079 | 5888 | 5192 | 3426 | 2295 | 8721 | 29601 |
> | Gowalla | 30044 | 5329 | 3387 | 2426 | 1834 | 7801 | 50821 |

---

### Official Review · Reviewer_3y1q · 2022-10-24

**Confidence:** 4
**Correctness:** 4
**Technical Novelty And Significance:** 4
**Empirical Novelty And Significance:** 3
**Recommendation:** 8

**Clarity, Quality, Novelty And Reproducibility:**

Clarity: The paper is mostly clear and easy to follow. However, I noticed a couple of undefined notations in Eq(8) on page 4: p_s, n_s, S, \hat{y_{i, p_s}, \hat{y_{j, n_s}.

Quality and Novelty: The proposed idea of using SVD decomposition could potentially be a good addition to the choice of augmentation methods. However, as I mentioned above, the comparison with several important papers from graph contrastive learning (GCL) literature is missing. Additionally, in one of the recent empirical studies on GCL literature [1], it was observed that the local-local contrastive learning approaches typically outperform local-global contrastive methods. Given that the current paper falls in the latter category, it might be even more insightful to compare Light GCL with local-local contrastive methods.

Reproducibility: The results can be reproduced with the help of provided code link.


References:
[1] Zhu, Yanqiao, et al. "An empirical study of graph contrastive learning." arXiv preprint arXiv:2109.01116 (2021).

[2] Velickovic, Petar, et al. "Deep Graph Infomax." ICLR (Poster) 2.3 (2019): 4.

[3] Hassani, Kaveh, and Amir Hosein Khasahmadi. "Contrastive multi-view representation learning on graphs." International Conference on Machine Learning. PMLR, 2020.

[4]  Peng, Zhen, et al. "Graph representation learning via graphical mutual information maximization." Proceedings of The Web Conference 2020. 2020.

[5] You, Yuning, et al. "Graph contrastive learning with augmentations." Advances in Neural Information Processing Systems 33 (2020): 5812-5823.

[6] Zhu, Yanqiao, et al. "Deep graph contrastive representation learning." arXiv preprint arXiv:2006.04131 (2020).

[7] Zhu, Yanqiao, et al. "Graph contrastive learning with adaptive augmentation." Proceedings of the Web Conference 2021. 2021.


**Strength And Weaknesses:**

Strengths:
1. The idea of using low-rank SVD to produce augmenting view of the original graph looks neat and well-aligned with the intuition of preserving global structure in the original graph.

2. The results indicate significant improvement in recall@k and ndcg@k with respect to all the baselines on all the datasets. Additionally LightGCL is able to produce good performance on sparse users with respect to the two baselines, SimGCL and HCCF.

3. LightGCL is computationally more efficient to train than the above baselines.


Weaknesses:
1. I did not understand why related work is missing several relevant papers from the domain of Graph Contrastive Learning [1-7]. Many of these papers are well-cited and published at top-tier conference.
2. The methods from above papers are also not included in the comparison study in the evaluation section.
3. The experiments on balance between over-smoothing and over-uniformity requires a bit more convincing. Ideally, if authors could demonstrate some alignment between clusters of entities learnt by the Light GCL and the domain-based clusters, that would be even better.

**Summary Of The Paper:**

The paper presents a new SVD-based Graph Contrastive Learning paradigm called LightGCL to learn effective representations of nodes in a user-item interaction graph from unlabeled data that are eventually used to predict users' preferences in downstream recommendation tasks. The key idea is  in the augmentation step where an alternate low-rank view of the original graph is generated by applying SVD on the original graph. The embeddings of each node from two graphs are trained to be similar by minimizing Info Noise Contrastive Loss along with the prediction loss on user-item interactions.

The proposed approach LightGCL is evaluated against SOTA methods from the various paradigms including: MLP-based Collaborative Filtering (CF), GNN-based CF, Disentangled Graph CF,  Hypergraph-based CF, and self-supervised recsys on five real-world datasets on recall@N and NDCG@N. The experiments indicate superior performance of LightGCL over all the above baselines. Additionally. authors demonstrate LightGCL outperforming a couple of baselines (HCCF and SimGCL) on less popular/interacting segments of items and users.

**Summary Of The Review:**

The paper presents an interesting approach for generating alternate view of the graph that could be a useful addition to Graph Contrastive Learning literature. The results look impressive overall, however the comparisons with some of the important papers are missing. EDIT: I am satisfied with author's follow-up edits to the paper and they reasonably address most of my comments. In light of the same, I have updated my rating for the paper.

---

> ### Author Response · Authors · 2022-11-18
> **Response to Reviewer 3y1q (1/2)**
>
> We are grateful for your detailed feedback and suggestions! Please find our updates and modifications in response to your feedback below. If our responses resolve your concerns, we genuinely hope that you can increase your rating score.
>
> ### **1) Discussion and comparison with suggested relevant works.**
>
> **Response**: Thank you for pointing out some relevant works. Regarding the suggested related work and baselines, we have enriched them in our updated paper as follows:
>
> **i)** In our updated manuscript, we have added all the relevant papers you suggested into the related work section for discussion as follows.
>
> > **Self-Supervised Learning on Graphs**. Recently, self-supervised learning (SSL) has advanced the graph learning paradigm to enhance node representation from unlabeled graph data (Zhu et al., 2021a; Velickovic et al., 2019; Hassani & Khasahmadi, 2020; Peng et al., 2020; Zhu et al., 2020b; 2021c). For example, to improve the predictive SSL paradigm, AutoSSL (Jin et al., 2022) automatically combines multiple pretext tasks for augmentation. Towards the line of contrastive SSL, GCA (Zhu et al., 2021c) performs both topology-level and attribute-level data augmentation for contrastive view generation. In this method, important edges and features will be identified for adaptive augmentation. GraphCL (You et al., 2020b) generates correlated graph representation views using various augmentation strategies, such as node/edge perturbation and attribute masking.
>
> **ii)** To further address the reviewer's comment, we have added three of the models in these papers into our performance comparisons: GraphCL, GRACE, and GCA. In particular, GCA has been added to the overall performance comparison (Table 1). Due to space limit, we put the other two along with some other models into Table 5 in Appendix B.
>
> **iii)** In addition, we included three other up-to-date GCL baselines into our comparisons: SimGRACE, SAIL, and AutoGCL.
>
> **iv)** Our model consistently outperformed the newly added baselines. Please refer to Table 1 and Table 5 in our updated paper for the details. We also attach the results of the newly added baselines here for your easy reference.
>
>  | Data    | Metric | GraphCL | SAIL   | GRACE  | AutoGCL | SimGRACE | GCA    | **LightGCL** |
> |---------|--------|---------|--------|--------|---------|----------|--------|--------------|
> | Yelp    | R@20   | 0.0462  | 0.0471 | 0.055  | 0.0593  | 0.0603   | 0.0621 | **0.0793**   |
> |         | N@20   | 0.0401  | 0.0405 | 0.047  | 0.0494  | 0.0435   | 0.053  | **0.0668**   |
> |         | R@40   | 0.0764  | 0.0773 | 0.0917 | 0.1009  | 0.0989   | 0.1021 | **0.1292**   |
> |         | N@40   | 0.0511  | 0.0516 | 0.0605 | 0.065   | 0.0656   | 0.0677 | **0.0852**   |
> | Gowalla | R@20   | 0.0997  | 0.0999 | 0.0744 | 0.0832  | 0.0869   | 0.0896 | **0.1578**   |
> |         | N@20   | 0.0603  | 0.0602 | 0.0452 | 0.0484  | 0.0528   | 0.0537 | **0.0935**   |
> |         | R@40   | 0.1473  | 0.1472 | 0.1071 | 0.1291  | 0.1276   | 0.1322 | **0.2245**   |
> |         | N@40   | 0.0727  | 0.0725 | 0.0539 | 0.0605  | 0.0637   | 0.0651 | **0.1108**   |
> | ML-10M  | R@20   | 0.1659  | 0.1728 | 0.2107 | 0.2325  | 0.2254   | 0.2145 | **0.2613**   |
> |         | N@20   | 0.2038  | 0.2118 | 0.2476 | 0.2755  | 0.2686   | 0.2613 | **0.3106**   |
> |         | R@40   | 0.2560  | 0.2639 | 0.3075 | 0.3415  | 0.3295   | 0.3231 | **0.3799**   |
> |         | N@40   | 0.2250  | 0.2332 | 0.2711 | 0.3023  | 0.2939   | 0.2871 | **0.3387**   |
> | Amazon  | R@20   | 0.0360  | 0.0357 | 0.036  | 0.0325  | 0.0381   | 0.0309 | **0.0585**   |
> |         | N@20   | 0.0266  | 0.0264 | 0.0271 | 0.0241  | 0.0291   | 0.0238 | **0.0436**   |
> |         | R@40   | 0.0585  | 0.0581 | 0.0583 | 0.0553  | 0.0621   | 0.0498 | **0.0933**   |
> |         | N@40   | 0.0340  | 0.0338 | 0.0345 | 0.0318  | 0.0371   | 0.0301 | **0.0551**   |
> | Tmall   | R@20   | 0.0251  | 0.0254 | 0.0303 | 0.0312  | 0.0222   | 0.0373 | **0.0528**   |
> |         | N@20   | 0.0175  | 0.0177 | 0.021  | 0.0204  | 0.0152   | 0.0252 | **0.0361**   |
> |         | R@40   | 0.0416  | 0.0424 | 0.0505 | 0.0524  | 0.0367   | 0.0616 | **0.0852**   |
> |         | N@40   | 0.0233  | 0.0236 | 0.0281 | 0.0278  | 0.0203   | 0.0337 | **0.0473**   |
>
> ### **2) Performance comparison with local-local contrastive methods.**
>
> **Response**: Thanks for the feedback. Please note that the newly added baselines GCA and SimGRACE are local-local contrastive methods, and our model also achieved better performance than them. While local-local contrastive methods outperformed many local-global methods, it largely depends on the augmentation approach adopted by the methods. By injecting meaningful, non-random global information into the augmented view, the local-global methods could also achieve outstanding performances.

---

> > ### Author Response · Authors · 2022-11-18
> > **Response to Reviewer 3y1q (2/2)**
> >
> > ### **3) Definition of terms in Eq. 8.**
> >
> > **Response**: Thank you for this comment! To clarify: $p_s$ and $n_s$ refer to the positive and negative sampled items for a user; $S$ is the number of positive-negative item pairs for a user; $\hat y_{i,p_s}$ and $\hat y_{i,n_s}$ denote the predicted scores for a pair of positive and negative items of user $i$. We have added an explanation to these terms on Page 4 right before Eq. 8 in our revised paper version.
> >
> > ### **4) More discussions on the balance between over-smoothing and over-uniformity.**
> >
> > **Response**: Thanks for this constructive comment. In Section 4.5, we perform experiments to show the capability of our method in achieving the trade-off between the over-smoothing and over-uniformity of learned embeddings, to be reflective of both user-specific preference patterns and user-wide interaction dependencies. In practical recommendation scenarios, neither over-smoothed nor over-uniformed embeddings are quality representations for capturing complex user preference. Specifically, over-smoothed user embeddings become nearly indistinguishable, may not be able to reflect unique interests of individuals. Over-uniformed user embeddings can hardly capture the dependencies among different users, which is limited to preserving collaborative relations among similar users. Therefore, our LightGCL can generate a moderately dispersed embeddings distribution (as shown in Fig.4), which has good potential in pursing a nice trade-off between embedding over-smoothing and over-uniformity. For those CL-based baselines (e.g., SGL), due to the inherent nature of contrastive learning for pushing away large numbers of negative samples (different nodes), they generate over-uniformed embedding distributions. For non-SSL GNN-based methods (e.g., LightGCN), the over-smoothing phenomenon can be observed.

---

> > > ### Comment · Reviewer_3y1q · 2022-12-04
> > > **Convinced with the responses.**
> > >
> > > I appreciate authors for addressing my comments and providing the detailed responses. I will consider that in my final rating.

---

### Official Review · Reviewer_kpFd · 2022-10-30

**Confidence:** 5
**Correctness:** 3
**Technical Novelty And Significance:** 4
**Empirical Novelty And Significance:** 4
**Recommendation:** 8

**Clarity, Quality, Novelty And Reproducibility:**

This work has already released source code and data for experimental result reproducibility. Generally speaking, this work shows novelty from an interesting angle for enhancing graph contrastive learning-based recommender system. Both effectiveness and efficiency are guaranteed by the new model.

**Strength And Weaknesses:**

1.Reasonable and well-motivated methodology design to advance the GCL-based recommender systems from a novel perspective of decomposition-based augmentation.
2. Sufficient and appropriate state-of-the-art methods (e.g., some recent models SGL, HCCF, SimGCL) are adopted for performance comparison.
3. Extensive experiments to demonstrate the effectiveness of the newly proposed solution from various aspects, including model robustness against data sparsity and long-tail issue.
4. Both the superior performance and lower model complexity are achieved by the new method compared with strong state-of-the-arts.

Please find my comments below:

Although the proposed method is simple yet effective from the experimental perspective, the authors are encouraged to discuss the effectiveness from the theoretical perspectives.

Besides, in performance comparison, I notice that different types of methods are considered as baselines, including representative GNN-based CF models (GCCF and LightGCN), and disentangled graph CF approach-DGCF. Additionally, recently proposed SSL-enhanced recommender systems SGL, SHT, HCCF are also included. For those hypergraph-enhanced approaches, the self-supervised information source mainly comes from information aggregation over hyperedges. The settings of hypergraph structures in those methods can be further described, which may be helpful to understand the effectiveness of SSL-based graph augmentation.


**Summary Of The Paper:**

This paper tackles the emerging topic of graph contrastive learning in recommender system, by proposing a lightweight and effective method. Particularly, the augmented view is automatically generated based on singular value decomposition technique, which is interesting and novel to me. With the augmented user-item graph structures, new contrastive self-supervision signals are derived. By doing so, only one additional augmentation view is needed for contrastive learning, further improving the model efficiency as justified in this work. Extensive experiments are conducted on several benchmark datasets to validate the effectiveness and superiority of the new approach.

**Summary Of The Review:**

Reasonable and well-motivated methodology. Comprehensive experiments are conducted for performance validation. The detailed setting descriptions of hypergraph-enhanced baselines can be provided, and theoretical effectiveness is also encouraged to supplement.

---

> ### Author Response · Authors · 2022-11-18
> **Response to Reviewer kpFd**
>
> Thank you very much for your helpful suggestions! Here are our responses to your comments:
>
> ### **1) Theoretical discussions.**
>
> **Response**: Thank you for your suggestion! We added theoretical analyses to show that our local-global CL (Eq. 7) benefits CF from two perspectives.
>
> Firstly, our CL is augmented to maximize the similarity between embeddings of potentially related nodes, based on the SVD-based global relation learning. Specifically, for a node $v_j\in\mathcal{U}$, where $\mathcal U$ is the set containing $u_{i'}$ having $A_{i,i'}=0$ and $\hat A_{i,i'}\neq 0$, the embeddings are not updated by $s(z_{i,l}, g_{i,l})$ in the vanilla InfoNCE loss, as $v_j$ is not adjacent to $u_i$. In contrast, our local-global contrastive loss assigns the following gradients to the embeddings of $v_j$:
>
> $\partial s(z_{i,l}, g_{i,l}) / \partial g_{i,l-1} = \partial s\left( z_{i,l}, \sigma(\sum_{j\in\mathcal U} \alpha_{i,j} g_{j,l-1} + \sum_{A_{i,j'}\neq 0} \alpha_{i,j'} g_{j',l-1} )\right) / \partial g_{j,l-1}=\frac{z_{i,l}}{\|z_{i,l}\|\|g_{i,l}\|} \cdot \sigma'(\cdot) \cdot \alpha_{i,j}$
>
> where $\alpha_{i,j}$ denotes the normalization weight for node $u_i$ and $v_j$. In this way, the embeddings of nodes in $\mathcal{U}$ are also pulled close to $s_{i,l}$, which injects relatedness information learned by the SVD into the local-global CL optimization.
>
> Secondly, our local-gloabl CL regularizes the embeddings $z$ affected by inaccurate observed graph $A$ with adaptive gradients.
>
> Specifically, the alignment of our CL assigns gradients with adaptive magnitudes to nodes considering the similarity between the local view $A$ and the global view $\hat A$. The norm of gradients correlated to a negative sample $i'$ and an anchor node $i$ is as follows:
>
> $||c(i)||_2$
>
> $=||(\bar g_{i', l} - (\bar s_{i,l}^\top \bar g_{i', l}) \bar s_{i,l})^\top \exp(\bar z_{i,l}^\top \bar g_{i', l}/\tau) / \sum_{i'}\exp(\bar z_{i,l}^\top \bar g_{i', l}/\tau)||_2$
>
> $\propto \sqrt {1-(\bar s_{i,l}^\top \bar g_{i',l})^2 } \exp(\bar s_{i,l}^\top \bar g_{i', l} / \tau)$
>
> where $\bar g_{i',l}$ and $\bar s_{i,l}$ are normalized embeddings. The last term grows substantially when $s_{i,l}$ and $g_{i',l}$ are similar to each other. This adaptive magnitude adjustment enables our CL to better discriminate the hard negatives by referring to the learned global relations in $\hat A$.
>
> A theoretical analysis section is also added to Appendix C of our updated paper.
>
> ### **2) Descriptions of hypergraph structure settings in hypergraph-enhanced recommender baselines.**
>
> **Response:** For the hypergraph-based baseline approaches HCCF and SHT, they generate hypergraph structures over user or item nodes to establish their high-order connections. To be consistent with the settings in their original papers, the number of hyperedges in HCCF and SHT is set as 128 in our experiments. The node-hyperedge message passing process is guided by the learned connection strength. Our experiments show that LightGCL method significantly outperforms those baselines, which demonstrates the effectiveness of our SVD-enhanced graph contrastive learning with injecting global collaborative signals.
>
> The setting descriptions for the hyper-graph configurations of HCCF and SHT in our experiments are added to the corresponding sections in Appendix A of our updated manuscript.

---

### Author Response · Authors · 2022-11-18
**General Response**

We are grateful to all the reviewers for their positive feedback and insightful suggestions! We really appreciate your kind consideration.

We are glad to hear that the reviewers consider our work "reasonable and well-motivated" [Reviewer kpFd], and acknowledge that the paper "tackles important challenges" [Reviewer rCJG] of GCL recommenders. We also sincerely appreciate the suggestions they provided, such as enriching related work [Reviewer 3y1q and k7CA], further clarifying the advantages of our model [Reviewer rCJG and k7CA], adding more theoretical discussions [Reviewer kpFd], enriching ablation study [Reviewer k7CA] and clarifying terms and configurations [Reviewer 3y1q and rCJG].

We provide specific detailed responses to each reviewer below. We also updated our manuscript to address the reviewer's concerns. The major changes are listed below, and also mentioned in the corresponding responses to each reviewer.

 - Related Work (Section 2): We enriched this part to include the relevant works suggested by Reviewer 3y1q and k7CA.
 - Methodogy (Section 3): We added an overall description of the model structure in Fig. 1 (suggested by Reviewer k7CA), and further defined unclear terms in Eq. 8 (suggested by Reviewer 3y1q).
 - Performance Comparisons (Section 4.1.2; Table 1 & 5; Appendix A & B): We included **six** additional baseline methods as suggested by Reviewer 3y1q and k7CA. Two of them are added to the overall table (Table 1). Due to space limits, the others are put to Table 5 of Appendix B. The baseline descriptions in Section 4.1.2 and Appendix A are also updated correspondingly.
 - Statistical Significance (Table 1): We added the *p-values* and *improvement ratios* as suggested by Reviewer k7CA.
 - Popularity Bias (Section 4.4): We appended an explanation of the different trends on *Yelp* and *Gowalla* datasets as suggested by Reviewer rCJG.
 - Ablation Study (Section 4.6): We added a new variant to the ablation study as suggested by Reviewer k7CA.
 - Theoretical Analysis (Appendix C): We present a theoretical analysis in Appendix C as suggested by Reviewer kpFd.

---

### Decision · Program_Chairs · 2023-01-20

**Decision:**

Accept: notable-top-25%

**Justification For Why Not Higher Score:**

While this is a good paper, I do not think it is groundbreaking enough to be considered as an oral presentation.

**Justification For Why Not Lower Score:**

The paper is solid in that there is a good motivation (deal with the biases of contrastive noise perturbation) and a novel way to implement it. The resulting approach is meaningful, lightweight and interesting from both the theoretical and applied perspective.

**Metareview: Summary, Strengths And Weaknesses:**

This work proposes a novel approach to achieving graph contrastive learning for recommendation problems, especially when there are sparsity issues or popularity biases. The key idea is to generate a low-rank representation of the graph using SVD which can be used in the contrastive augmentation step.

The main idea of this paper is novel and simple. The authors describe and motivate it well. Furthermore, this approach seems to be quite effective. This has been tested in appropriate datasets for recommender systems and also against a range of baselines - including new baselines which were suggested by reviewers during the review period. In fact, this method is very consistent in outperforming the other baselines and the code to reproduce the experiments is already provided by the authors.

Overall, this is a very interesting and well-written paper which should readily be of interest to practitioners but also to researchers who should be able to build on that for further advances.


**Note From Pc:**

if the above contains the word "oral" or "spotlight" please see: "oral" presentation means -> notable-top-5% and "spotlight" means -> notable-top-25%. As stated in our emails, we are disassociating presentation type from AC recommendations